# Effects of Olfactory Mucosa Stem/Stromal Cell and Olfactory Ensheating Cells Secretome on Peripheral Nerve Regeneration

**DOI:** 10.3390/biom12060818

**Published:** 2022-06-11

**Authors:** Rui D. Alvites, Mariana V. Branquinho, Ana C. Sousa, Bruna Lopes, Patrícia Sousa, Justina Prada, Isabel Pires, Giulia Ronchi, Stefania Raimondo, Ana L. Luís, Stefano Geuna, Artur Severo P. Varejão, Ana Colette Maurício

**Affiliations:** 1Centro de Estudos de Ciência Animal (CECA), Instituto de Ciências, Tecnologias e Agroambiente da Universidade do Porto (ICETA), Rua D. Manuel II, Apartado 55142, 4051-401 Porto, Portugal; ruialvites@hotmail.com (R.D.A.); mevieira@icbas.up.pt (M.V.B.); anacatarinasoaressousa@hotmail.com (A.C.S.); brunisabel95@gmail.com (B.L.); pfrfs_10@hotmail.com (P.S.); alluis@icbas.up.pt (A.L.L.); 2Departamento de Clínicas Veterinárias, Instituto de Ciências Biomédicas de Abel Salazar (ICBAS), Universidade do Porto (UP), Rua de Jorge Viterbo Ferreira, nº 228, 4050-313 Porto, Portugal; 3Associate Laboratory for Animal and Veterinary Science (AL4AnimalS), 5000-801 Vila Real, Portugal; jprada@utad.pt (J.P.); ipires@utad.pt (I.P.); avarejao@utad.pt (A.S.P.V.); 4Centro de Ciência Animal e Veterinária (CECAV), Universidade de Trás-os-Montes e Alto Douro (UTAD), Quinta de Prados, 5001-801 Vila Real, Portugal; 5Departamento de Ciências Veterinárias, Universidade de Trás-os-Montes e Alto Douro (UTAD), Quinta de Prados, 5001-801 Vila Real, Portugal; 6Department of Clinical and Biological Sciences, and Cavalieri Ottolenghi Neuroscience Institute, University of Turin, Regione Gonzole 10, 10043 Orbassano, Italy; giulia.ronchi@unito.it (G.R.); stefania.raimondo@unito.it (S.R.); stefano.geuna@unito.it (S.G.)

**Keywords:** peripheral nerve injury, peripheral nerve regeneration, sciatic nerve, olfactory mucosa mesenchymal stem/stromal cells, olfactory ensheating cells, secretome, conditioned medium, nerve guidance conduit, tibial cranial muscle, rat

## Abstract

Cell secretome has been explored as a cell-free technique with high scientific and medical interest for Regenerative Medicine. In this work, the secretome produced and collected from Olfactory Mucosa Mesenchymal Stem Cells and Olfactory Ensheating Cells was analyzed and therapeutically applied to promote peripheral nerve regeneration. The analysis of the conditioned medium revealed the production and secretion of several factors with immunomodulatory functions, capable of intervening beneficially in the phases of nerve regeneration. Subsequently, the conditioned medium was applied to sciatic nerves of rats after neurotmesis, using Reaxon^®^ as tube-guides. Over 20 weeks, the animals were subjected to periodic functional assessments, and after this period, the sciatic nerves and cranial tibial muscles were evaluated stereologically and histomorphometrically, respectively. The results obtained allowed to confirm the beneficial effects resulting from the application of this therapeutic combination. The administration of conditioned medium from Olfactory Mucosal Mesenchymal Stem Cells led to the best results in motor performance, sensory recovery, and gait patterns. Stereological and histomorphometric evaluation also revealed the ability of this therapeutic combination to promote nervous and muscular histologic reorganization during the regenerative process. The therapeutic combination discussed in this work shows promising results and should be further explored to clarify irregularities found in the outcomes and to allow establishing the use of cell secretome as a new therapeutic field applied in the treatment of peripheral nerves after injury.

## 1. Introduction

Peripheral Nerve Injuries (**PNIs**) are common clinical phenomena with a multifactorial origin that represent a medical challenge around the world. The anatomical characteristics of the peripheral nerve and its often superficial and exposed position make this type of injury a frequent occurrence. As it is not a life-threatening condition, these injuries are often underdiagnosed and undervalued, even though they are often accompanied by permanent motor, sensory, nociceptive, and autonomic changes that translate into long-term disabilities, major medical investments, and permanent loss of function in the extremities. Full functional recovery is difficult and rare, particularly in more severe injuries and with delayed interventions [1]. Thus, there has been a medical and scientific investment in the search for new and effective therapeutic approaches that allow overcoming the identified limitations, particularly involving tissue engineering, cell-based therapies, and the improvement of microsurgical techniques, in innovative therapies in the field of regenerative medicine [2,3].

After PNIs, the Wallerian degeneration phenomena and the release of pro-inflammatory cytokines and growth factors allow a well-documented spontaneous nerve regeneration, although its limits are still unclear [4]. One of the questions that remains is why these physiological procedures often become out of control and lead to the development of glial scars, intense fibrosis and neuromas, and the occurrence of misdirecting and misalignment of nerve fibers under regeneration, which often translates into persistent neuropathic pain [5]. The understanding that the success or failure of the regenerative process after PNIs is dependent on the local inflammatory balance led to the search for therapeutic approaches capable of performing immunomodulation. Mesenchymal Stem Cells (**MSCs**) have been widely studied in this context, as therapeutic systems with autologous potential are capable of stimulating the regeneration of central and peripheral nerve tissue through the secretion of neurotrophic growth factors, accelerating axonal growth, supporting the survival of neurons, and inducing the differentiation of other cells in the neuroglial lineages, demonstrating promising results both in vitro and in vivo [6,7].

Among the mechanisms of action described, MSCs are able to replace damaged and lost tissues and cells, to differentiate into target tissue cells, to secrete bioactive factors such as homing and signaling molecules, and also to directly influence the remyelination processes, essential for the functional and structural recovery of the injured nerve. In addition, MSCs are considered to be immune-privileged, almost never causing immune and rejection reactions after allogeneic transplantation, greatly increasing their range of applicability [8].

MSCs have already been isolated, expanded, and characterized from niches in virtually all body tissues, and the olfactory mucosa (**OM**) MSCs (**OM-MSCs**) are a type of MSCs originating in the ectoderm and neural crest that only in recent years have begun to receive attention in this context. With their peripheral location in the nasal cavity that allows easy access and collection of OM, OM-MSCs have already been isolated from the lamina propria in several species [9]. These studies resulted in the establishment of numerous advantageous characteristics that make OM-MSCs therapeutic elements with high clinical potential, namely their wide distribution within the nasal cavity and ethmoidal labyrinth that allow an easy collection both ante-mortem and post-mortem, their chromosomal stability, and the ability to self-renew for long periods of time without apparent influence of donor’s age. There are few works involving these cells and their application, and our group has made efforts to study in greater depth their molecular characteristics and potential for clinical application. After establishing a protocol that allows the collection of OM in the rat model, quickly, free of contamination, and immediately after euthanasia, it was possible to isolate the cells in vitro and proceed with their complete characterization [10]. In this exploratory work, it was possible to confirm the minimum criteria for classification of MSCs established by the International Society for Cellular Therapy (**ISCT**) [11], namely their plastic-adherence in culture, the ability to tri-differentiate, and the expression of typical surface markers. The expression of neural cell-related genes, their chromosomal stability, ability to form colonies, and the cellular kinetics advantageous for their later application in vivo were also confirmed. Additionally, a preliminary study of its conditioned medium (**CM**) was carried out, identifying the production and secretion of several factors with immunomodulatory potential [12].

There are few studies in which OM-MSCs were applied in vivo, but they have shown their therapeutic potential in the treatment of neurodegenerative diseases [13], spinal cord trauma [14], and immune-mediated diseases [15]. Its application in treatments after PNI has also been explored [16]. Following the extensive characterization work around OM-MSCs, our research group proceeded to apply these cells in a neurotmesis rat model in combination with a commercially available chitosan nerve guidance conduit (**NGC**) (Reaxon^®^), this work representing the first application of this therapeutic combination in vivo. This methodology allowed to confirm the benefits of using OM-MSCs therapeutically and in combination with a Reaxon^®^ NGC (Medovent GmbH (Mainz, Germany)), maximizing the benefits of both individual therapeutic approaches. However, the results obtained were not free of irregularities, with the different therapeutic groups obtaining variable performances in functional and histomorphometric recovery evaluation assays, not allowing to unequivocally establish this therapeutic approach as ideal for further translation studies [17].

The observation of irregular results after the application of MSCs in vivo, despite all the advantageous characteristics identified in the in vitro studies, is not a rare occurrence [18]. It is known that in order for MSCs to exert their therapeutic potential per se, they must reach the site of injury and survive the inflammatory environment that has been established there [19]. This cell survival is dependent on the site of cell administration (at the site of injury or systemically), the rate of injection, and the density of cells administered. Recent studies, however, demonstrate that, regardless the method of administration, only a small percentage of the administered cells manage to survive after transplantation. The main mechanisms of loss are systemic dilution and cell death after exposure to the hostile environment around the lesion [8]. Although the application of NGCs and the administration of MSCs inside their lumen allow to reduce the occurrence of these deleterious phenomena, the mechanical microaggressions that can occur during the administration of the cells can also lead to their death and even to the development of neuromas in the nerve under regeneration [20]. Thus, the need remains to find effective therapeutic solutions for application after PNI that are able to overcome the limitation of cell application.

The fact that it is known that after cell transplantation, the proportion of cells that effectively survive and integrate the site of injury is reduced, but it is still possible to observe therapeutic benefits after their administration, has raised the possibility that MSCs act in ways other than direct cell action. Several recent studies have demonstrated that MSCs can act and influence the surrounding environment not only through cell-to-cell interactions but also through paracrine signaling processes [21]. Through these mechanisms, and after being activated after contact with inflammatory biofactors, MSCs are able to produce and secrete a set of soluble molecules, including growth factors, cytokines, chemokine, growth factors, and extracellular vesicles that directly influence the inflammatory and regenerative processes in a way similar to the administered cells [8]. This fact creates an alternative therapeutic approach that allows using all the medicinal benefits of MSCs without the limitations associated with their direct administration. The use of cell-free strategies has clear advantages over the traditional method, namely ensuring a safer administration with no risks associated with the administration of cells, in addition to an easier and more practical handling and storage [22].

The set of soluble factors produced and secreted by a cell in a given culture condition is called a secretome [23]. The secretome collected under a certain culture condition can be called CM. For instance, through in vitro priming methods, it is possible, by modifying the culture conditions, to indirectly influence the expression of MSCs’ genes and also their secretion products, thus manipulating the immunomodulatory behavior of the cell and allowing the production of a secretome oriented towards the regeneration of a certain tissue. A common priming approach is to manipulate the cell secretome by supplementing the culture medium with soluble factors such as cytokines [8].

Previously, our research group carried out an exploratory work in which the CM of OM-MSCs was analyzed, allowing to identify the production and secretion of immunomodulatory factors with the potential to influence the regenerative process [12]. In this work, in addition to performing an additional analysis of the CM of OM-MSCs in different passages, we tested the hypothesis that the application of CM obtained from the conditioning of OM-MSCs, used in combination with Reaxon^®^ NGCs, may enhance the histomorphometric and functional regeneration of the rat’s sciatic nerve after neurotmesis in a similar way to the therapeutic application of cells per se. Additionally, CM from rat Olfactory Ensheating Cells (**OECs**), a type of specialized OM glial cells that have previously demonstrated the ability to promote regeneration in the peripheral nervous system [24], was also analyzed and used therapeutically for efficacy comparison.

## 2. Materials and Methods

### 2.1. Cells Conditioned Medium Analysis—Secretome

OM-MSCs were harvested from the rat olfactory mucosa and maintained in culture as previously described [17]. OECs were cultured according to the manufacturer’s instructions (Rolf B1.T, Merck KGaA^®^, Darmstadt, Germany). Both cell types were subjected to an analysis of their CM in order to identify cytokines and chemokines secreted after conditioning. To compare changes in secretory capacity over time, cells in two different passages (P4 and P7) were subjected to conditioning.

When in culture, after reaching a confluence of around 70–80%, the culture medium was removed, and the culture flasks were gently washed with Dulbecco’s phosphate-buffered saline (DPBS) two to three times (2 to 3×). Then, the culture flasks were further washed two to three times with the basal culture medium of each cell type, without any supplementation. To begin the conditioning, non-supplemented DMEM/F12 GlutaMAX™ (10565018, Gibco^®^, Thermo Fisher Scientific^®^, Waltham, MA, USA) culture medium was added to the culture flasks, which were then incubated under standard conditions. The culture medium rich in factors secreted by the cells (CM) was collected after 24 and 48 h.

The collected CM was then concentrated five times (5×). After collection, it was centrifuged for 10 min at 1600 rpm, its supernatant collected and filtered with a 0.2 μm Syringe filter (Filtropur S, PES, Sarstedt^®,^ Nümbrecht, Germany). For the concentration procedure, Pierce™ Protein Concentrator, 3k MWCO, 5–20 mL tubes (88525, Thermo Scientific^®^, Waltham, MA, USA) were used. Initially, the concentrators were sterilized following the manufacturer’s instructions. Briefly, the upper compartment of each concentrator tube was filled with 70% ethanol (*v*/*v*) and centrifuged at 300× *g* for 10 min. At the end of the centrifuge, the ethanol was discarded, and the same procedure was carried out with DPBS. Each concentrator tube was subjected to two such centrifugation cycles, followed by a 10-min period in the laminar flow hood for complete drying. Finally, the upper compartment of the concentrator tubes was filled with plain CM (1× concentration) and subjected to new centrifugation cycle, under the conditions described above, for the number of cycles necessary to obtain the desired CM concentration (5×).

The concentrated CM was stored at −20 °C and subsequently subjected to a Multiplexing LASER Bead analysis (Eve Technologies, Calgary, AB, Canada) to identify a set of biomarkers present in the Rat Cytokine/Chemokine 27-Plex Discovery Assay^®^ Array (RD27). The list of searched biomarkers includes Eotaxin, Epidermal Growth Factor (**EGF**), Fractalkine, Interferon Gamma (**IFN-γ**), Interleukins (**IL**) IL-1α, IL-1β, IL-2, IL-4, IL-5, IL-6, IL-10, IL-12p70, IL- 13, IL-17A, IL-18, IP-10, Human Growth-regulated oncogene/Keratinocyte Chemoattractant (**GRO/KC**), Tumor Necrosis Factor Alpha (**TNF-α**), Granulocyte Colony Stimulating Factor (**G-CSF**), Granulocyte-macrophage Colony-stimulating Factor (**GM-CSF**), Monocyte Chemoattractant Protein-1 (**MCP-1**), Leptin, LIX, Macrophage Inflammatory Protein (**MIP**) MIP-1α, MIP-2, Regulated on Activation, Normal T Cell Expressed and Secreted (**RANTES**), and vascular endothelial growth factor (**VEGF**). All samples were analyzed in duplicate.

### 2.2. In Vivo Assays

Assays including animal’s manipulation or intervention were previously approved by the Organism Responsible for Animal Welfare (ORBEA) of the Abel Salazar Institute for Biomedical Sciences (ICBAS) from the University of Porto (UP) (project 209/2017) and by the Veterinary Authorities of Portugal (DGAV) (DGAV project: 2018-07-11014510). All phases were designed in accordance with the assumptions contained in Directive 2010/63/EU of the European Parliament and its transposition into the Portuguese DL 113/2013, and based on the OECD Guidance Document on the Recognition, Assessment and Use of Clinical Signs the Humane Endpoints for Experimental Animals Used in Safety Evaluation (2000). All measures have been taken to ensure animal welfare and to avoid unnecessary discomfort and pain, considering all human endpoints for animal suffering and distress.

Thirty rats (*Rattus norvegicus*), Sasco Sprague Dawley Breed, 8- to 9-week-old, and approximately 250–300 g BW (Charles River, Barcelona, Spain) were used in this work. We opted for the exclusive use of males, thus avoiding the interference of hormonal variations resulting from the different reproductive phases observed in females in the daily behavior. After reception, animals underwent a two-week acclimatization period. The animals were housed with a density of two to three animals per cage, with ad libitum access to water and chow and environmental enrichment, allowing the manifestation of normal physiological behaviors and activities under standard laboratory conditions. Housing rooms were maintained at a temperature adapted to the species with 12–12 h light/dark cycles.

Prior to the surgical intervention, CM from OM-MSCs and OECs was produced, collected, and stored as previously described. P6 cells were used for both cell lines, and conditioning was maintained for 48 h.

#### 2.2.1. Experimental Design

Six therapeutic groups were established, distributing the animals subjected to a neurotmesis lesion (Figure 1): group 1—Uninjured control (**UC**) (*n* = 28); group 2—End-to-end suture (**EtE**) (*n* = 5); group 3—Suture of the nerve ends to the Reaxon^®^ nerve conduit and administration of OM-MSCs CM (300 µL) (**CMOM**) (*n* = 6); group 4—EtE, wrapping with Reaxon^®^ nerve conduit and administration of OM-MSCs CM (300 µL) (**ECMOM**) (*n* = 6); group 5—Suture of the nerve ends to the Reaxon^®^ nerve conduit and administration of OECs CM (300 µL) (**CMOEC**) (*n* = 5); group 6—EtE, wrapping with Reaxon^®^ nerve conduit and administration of OECs CM (300 µL) (**ECMOEC**) (*n* = 6). Reaxon^®^ NGCs with 15 mm long and 2.1 mm internal diameter were applied. Neurotmesis injury was obtained through a complete transection of the sciatic nerve in the animals’ right lateral limb, with the left contralateral limb being used as UC. Primary outcomes included the assessment of motor, nociceptive, and behavioral evolution over the study period; the secondary outcomes were based on the stereological evaluation of the sciatic nerve after harvesting and on the determination of the mass and histomorphometric characteristics of the cranial tibialis muscle (*M. tibialis cranialis*) as an effector muscle of the sciatic nerve.

#### 2.2.2. Surgical Procedures

The surgical procedures applied in this work were based on the techniques previously used and described [17,25]. The specificities and adaptations are described below.

##### Sciatic Nerve Repair Model

The anesthetic protocol used consisted of a combination of Xylazine/Ketamine (Rompun^®^ (Bayer Portugal^®^, Carnaxide, Portugal)/Imalgene^®^ (Merial^®^, Boehringer Ingelheim Portugal^®^, Lisboa, Portugal) (1000^®^; 1.25 mg/9 mg per 100 g BW)), administered intraperitoneally. Once anesthetic induction was confirmed, animals were placed in left lateral recumbency for exposure of the right limb, which was prepared for surgery (trichotomy and asepsis). The procedures were conducted under an M-650 operating microscope (Leica Microsystems, Wetzlar, Germany). Surgical access was made through a surgical incision starting at the greater trochanters of the femur and extending distally, up to the mid half of the thigh. The subcutaneous tissue was carefully debrided, allowing the physical separation of the vastus lateralis and biceps femoris muscles and the exposure of the sciatic nerve between them. The muscles were kept apart using a soft tissue retractor to increase the working field. After being separated from surrounding tissues, the sciatic nerve was carefully immobilized and subjected to neurotmesis (complete transection) using a straight microsurgical scissor. The transection was performed immediately proximal to the branching site of the sciatic nerve in its main branches, the common peroneal nerve and tibial nerve. The contralateral left nerve was considered as a control and part of the UC group.

The transectioned nerves were subjected to one of the described therapeutic interventions. In the EtE group, the nerve ends were placed and aligned in order to maintain a correct anatomical orientation and approximated to maintain a minimal gap between the nerve ends. Two to four simple interrupted epineural microsutures using 7/0 monofilament nylon material were applied between the two nerve endings to ensure maintenance of anatomical alignment and avoid posterior rotations. In the CMOM group, approximately 3 mm of the proximal and distal nerve ends were introduced into the lumen of the Reaxon^®^ NGCs, and fixed to the guide tube with two to four microsutures using 7/0 monofilament nylon material in order to keep the nerve as aligned and oriented as possible and with a gap of 9–10 mm between the nerve tops. Then, the OM-MSCs CM was injected inside the NGC, filling its internal diameter in direct contact with the injured nerve. In the ECMOM group, a tension-free suture was applied as in the EtE group, followed by wrapping the nerve and the suture site with a Reaxon^®^ NGC with a longitudinal incision and filling the internal diameter of the NGC with OM-MSCs CM. The nerve + NGC set was then carefully accommodated between neighboring muscles to avoid CM licking and subsequent displacement of the biomaterial. In the CMOEC and ECMOEC groups, the procedures were the same as in the CMOM and ECMOM groups, with the alternative application of OECs CM. Once the therapeutic intervention was completed, simple interrupted sutures with 4-0 absorbable material were applied to simultaneously close the muscular, subcutaneous and cutaneous layers. A deterrent substance was applied to the right paw of the intervened animals to avoid autotomy phenomena secondary to denervation. The animals were carefully monitored during anesthetic recovery, and later transferred to the original cages and social groups once the return to normal activity and behavior was confirmed. In the five days following the surgical interventions, the animals were medicated with carprofen (2–5 mg/kg SC QD), and throughout the entire study period, they were regularly evaluated and monitored to identify any signs of contractures, autotomies, wounds, and skin infections.

#### 2.2.3. Functional Assessment

After the surgical interventions and application of therapeutic options, the animals were subjected to a set of evaluation tests in order to assess the functional progression and recovery associated with the regenerative process. All animals were tested before surgery (Week 0) to establish functional and behavioral baseline values in healthy animals. The animals were then evaluated one week and two weeks after neurotmesis for hyperacute assessment, and every two weeks thereafter until week 20 post-surgery. The set of tests of all sessions were always performed by the same operator, with previous experience in the used methods, in order to avoid interindividual variations. In each session, the order of the evaluated animals was random and the animals were selected blindly, to avoid that the animal’s identity influenced the operator’s evaluation. All measures were taken to ensure that the assessments were carried out in a calm and stress-free environment so as not to condition the animals’ performance.

##### Motor Performance

Motor performance was determined using the extensor postural thrust (**EPT**) test, as previously described [17,25]. In this test, the animal is wrapped in a protective cloth, allowing exposure of the head and limb to be tested. The animal is then suspended over a digital weighting machine (model TM 560; Gibertini, Milan, Italy) and the body lowered towards it. By keeping the head exposed, the animal can visually anticipate contact with the weighting machine, voluntarily extending the exposed limb. The approach must be made in such a way that the contact with the weighting machine is made with the distal metatarsals and digits. The force applied by the limb on the weighting machine is recorded in grams, for both the injured (**EEPT**) and healthy (**NEPT**) limbs. The quantification of the applied force is done in triplicate, and the average of the three records is the final value considered. Finally, the determined EEPT and NEPT values are introduced into the following equation [26] to determine the percentage of motor functional deficit:$$\%\;\mathrm{Modor}\;\mathrm{Deficit}=\frac{\mathrm{NEPT}-\mathrm{EEPT}}{\mathrm{NEPT}}\times 100$$

##### Nociceptive Function

The assessment of sensory function, namely the integrity of the nociceptive function, was determined using the withdrawal reflex latency (WRL) [17]. This test is based on thermal stimulation of the extremity of the intervened limb. The animal is again wrapped in a protective cloth with exposure of the limb to be tested, and then suspended over a heating plate at 56 °C (model 35-D, IITC Life Science Instruments, Woodland Hill, CA, USA). The paw of the limb to be tested is then placed on the surface of the heating plate, and the time, in seconds, it takes for the animal to withdraw the limb from the noxious/thermal stimulus is registered. This quantification is determined through three measurements on each limb, with an interval of two minutes between each measurement to avoid interference from sensitization. A healthy animal should remove the limb from the thermal stimulus in 4.3 s or less. In the case of animals with severe sensory deficits, 12 s is considered the maximum time of contact between the limb and the plate before thermal injuries occur, and if the animal does not remove the paw during this period, the operator must actively remove it to avoid burns [27]. Some care must be taken during the test. When placing the limb on the plate, it must be ensured that it is the lateral region of the palmar surface of the paw that contacts it, as this is the region innervated by the terminal branches of the sciatic nerve. The medial region is innervated by the saphenous nerve, a branch of the femoral nerve, and innervation and collateral sprouting of branches of this nerve to adjacent regions is a possible occurrence. Thus, proper contact of the lateral aspect with the plate is essential to obtain viable results [28]. In cases where muscle atrophy and contractures are observed, it may be necessary to apply slight manual compression on the paw to ensure contact of the palmar surface with the plate, but pressure should be avoided to prevent stimulation of mechanoreceptors that can lead to limb retraction, even without activation of the nociceptive reflex arc [29].

##### Walking Track Analysis

The gait pattern was analyzed through the determination of the sciatic functional index (**SFI**) and static sciatic index (**SSI**) using a video collection technique, as previously described [17]. To determine the SFI, a set is assembled, consisting of a transparent acrylic corridor, at the bottom of which is placed a dark shelter, and below which an image capture system is positioned. The animal is placed at the beginning of the corridor and encouraged to go to the shelter in search of refuge. When the animal passes, on its way, over the image capture system, it is possible to register the contact of the foot with the floor of the corridor. For each animal, three valid steps are considered, these being a step in which the entire foot is captured and paleness in the paw limits and support points are observed, as a result of the pressure against the floor of the corridor. The captured images are analyzed using an image processing software (Image-Pro Plus^®^ 6—Media Cybernetics, Inc., Rockville, MD, USA), measuring three distances in the obtained footprint both in the healthy and experimental paws: print length (**PL**), toe spread distance between toes 1 and 5 (**TS**), and toe spread between toes 2 and 4 (**ITS**). The mean values obtained are then used in the following formulas (N stands for “normal” and E stands for “experimental”):$$\mathrm{Toe}\;\mathrm{Spread}\;\mathrm{Factor}\;\left(\mathrm{TSF}\right)=\frac{\left(\mathrm{ETS}-\mathrm{NTS}\right)}{\mathrm{NTS}}$$
$$\mathrm{Intermediate}\;\mathrm{Toe}\;\mathrm{Spread}\;\mathrm{Factor}\;\left(\mathrm{ITF}\right)=\frac{\left(\mathrm{EIT}-\mathrm{NIT}\right)}{\mathrm{NIT}}$$

The calculated values are then introduced into the following formula:$$\mathrm{SFI}=\left(-38.3\times \;\mathrm{PLF}\right)+\left(109.5\times \;\mathrm{TSF}\right)+\left(13.3\times \;\mathrm{ITF}\right)-8.8\;$$

Derived from the SFI, the SSI considers the animal in a static position and does not include the PL values. In this test, the set is adapted, and the animal is placed inside an acrylic box, which in turn is placed inside the acrylic corridor. In this way, the two limbs of the animal are in contact with the floor of the box, allowing the capture of footprints using the image capture system placed below the corridor. The TS and ITS values are determined in both injured and healthy limbs, in triplicate, and the values entered in the following formulas:$$\mathrm{Toe}\;\mathrm{Spread}\;\mathrm{Factor}\;\left(\mathrm{TSF}\right)=\frac{\left(\mathrm{ETS}-\mathrm{NTS}\right)}{\mathrm{NTS}}$$
$$\mathrm{Intermediate}\;\mathrm{Toe}\;\mathrm{Spread}\;\mathrm{Factor}\;\left(\mathrm{ITF}\right)=\frac{\left(\mathrm{EIT}-\mathrm{NIT}\right)}{\mathrm{NIT}}$$

Finally, the values obtained are integrated into the following formula:$$SSI=\left(108.44\times TSF\right)+\left(31.85\times ITF\right)-5.49$$

The SFI and SSI values for healthy animals are 0 and −5, respectively, and −100 and −90 for animals with total impairment of the injured sciatic nerve, respectively [30].

### 2.3. Stereological and Histomorphometric Analysis

#### 2.3.1. Nerve Stereological Analysis

After the 20th week functional assessment, all animals were euthanized using general anesthesia followed by an overdose of Sodium Pentobarbital (Eutasil^®^, Ceva Saúde Animal^®^, Algés, Portugal), 200 mg/mL, 200 mg/kg b.w., intraperitoneally). After confirming the euthanasia, a surgical approach was performed identical to the one described above to expose the healthy and intervened sciatic nerves. Uninjured control nerves and 10 mm segments distal to the site of injury in the intervened nerves were collected and fixed for subsequent stereological evaluation by light and electron microscopy. Specifically, after exposure (Figure 2), the sciatic nerves were covered with a fixative solution (0.5% saccharose and 2.5% purified glutaraldehyde in 0.1 M Sorensen phosphate buffer at pH 7.4 and 4 °C) to promote nerve hardening and facilitate its collection and handling. The proximal end of each collected segment was properly identified and then immersed in the same fixative solution for 5 min in order to guarantee the maintenance of the nerve alignment and avoiding its coiling or bending. Then, the nerve segments were placed in recipients and immersed in the fixation solution and kept there for 6 to 8 h, to be then washed and immersed in a new washing solution with 1.5% saccharose in 0.1 M Sorensen phosphate buffer at pH 7.4 during 6 to 12 h. After the histological preparation as previously described [31,32,33], the histomorphometric evaluation was based on the parameters number of fibers (N), fiber density (N/mm^2^), axon diameter (d, μ), fiber diameter (D, μ), myelin thickness (M, μ), cross-sectional area (mm^2^), and the ratio d/D (g-ratio).

#### 2.3.2. Histomorphometric Muscle Analysis

The cranial tibial muscles were also collected simultaneously with the sciatic nerves for further histomorphometric evaluation and quantification of the degree of neurogenic atrophy. Immediately after harvesting, both the healthy and the intervened limb muscles were weighed to quantify the loss of muscle mass. The samples were then fixed in 4% buffered formaldehyde and subjected to processing for routine histopathological analysis (hematoxylin and eosin (H&E)). Consecutive 3 μm thick sections were obtained from the mid-belly region of the muscle, and these were prepared and suitably stained. Low magnification images (100×) were obtained with a Nikon^®^ (Nikon Corporation^®^, Tokyo, Japan) microscope connected to a Nikon^®^ digital camera DXM1200 and analyzed with ImageJ^®^ software (Rasband, W.S., ImageJ, U. S. National Institutes of Health, Bethesda, MD, USA) using an unbiased sampling procedure. Through the measurement of individual fibers, the parameters muscle fiber area and minimal Feret’s diameter (that is, minimum distance of parallel tangents at opposing borders of the muscle fiber) were calculated. Two operators with extensive experience in the applied methodology performed the measurements, blindly and randomly, on a minimum of 800 fibers for each study group.

#### 2.3.3. Statistical Analysis

Statistical analysis was performed using the GraphPad Prism software version 6.00 for Windows (GraphPad Software, La Jolla, CA, USA). Whenever appropriate, data and results were expressed as mean ± SEM. Comparisons between groups in the results of different tests are based on the application of a parametric test. A value of *p* < 0.05 was considered as statistically significant. Significance of the results is shown according to *p* values by the symbol *: * corresponds to 0.01 ≤ *p* < 0.05, ** to 0.001 ≤ *p* < 0.01, *** to 0.0001 ≤ *p* < 0.001, and **** to *p* < 0.0001.

## 3. Results

### 3.1. Conditioned Medium Analysis

The results of the OM-MSCs and OECs CM analysis can be found, respectively, in Figure 3 and Figure 4. The average concentration of each biomarker identified in the analyzed CM is found in Appendix A and the respective statistical differences in Appendix A. In the secretome of OM-MSCs, 10 biomarkers were identified: Frutalkine, GRO/KC, IFN-γ, IL-18β, IL-10, IP-10, LIX, MCP-1, Rantes, and VEGF. In the secretome of OECs, on the other hand, 19 biomarkers were identified: EGF, Frutalkine, G-CSF, GRO/KC, IFN-γ, IL-1α, IL-1β, IL-5, IL-10, IL-17A, IL-18, IP-10, Leptin, LIX, MCP-1, MCP-1α. MIP-2, Rantes, and VEGF.

In most biomarkers, they were identified in both P4 and P7. In cases where markers were only identified in one passage, it was usually at P7. In general, there is an increase in the concentration of biomarkers with increasing conditioning time, which is higher after 48 h than after 24 h of conditioning. On the other hand, at higher passages, higher concentrations of biomarkers are observed compared to P4. In OM-MSCs, the biomarkers Frutalkine, GRO/KC, LIX, and MCP-1 were only identified at passage P7; Rantes was only identified in P4, after 48 h of conditioning. In OECs, the biomarkers G-CSF, IL-17A, IL-18, Leptin, and MCP-1a were only identified in the CM of cells at P7, and there were no biomarkers that were only identified at P4.

### 3.2. Functional Assessment

#### 3.2.1. Motor Performance

The percentage of motor deficit (%) recorded over the 20 weeks of study can be seen in Figure 5. The complete values of motor deficit can be found in Appendix A, and the statistical differences observed in T20 in Appendix A.

Immediately after the neurotmesis injury, there was an evident loss of muscle strength in the hind limbs and a significant increase in the percentages of motor deficit in all intervened groups. One week after the injury, all groups showed high percentages of motor deficit when compared to the UC group (*p* < 0.001), and the EtE group show the highest percentage, with statistical differences with all the other therapeutic groups (*p* = 0.0065, with CMOM; *p* = 0.0001 with ECMOM; *p* = 0.0003 with CMOEC and *p* = 0.0074 with ECMOEC, respectively). Deficits gradually decreased over the study weeks, with the groups occupying the position with the best results alternating at each timepoint. From the tenth week until the last timepoint, there was a significant decrease in the motor deficit observed in all the therapeutic groups. At 20 weeks, the CMOM group showed the lowest value of motor deficit, standing out from the other therapeutic groups where the results were similar. In any case, the final percentage of motor deficit is still high for all intervention groups after 20 weeks, and all of them show statistically significant differences with the UC group (*p* < 0.0001). Additionally, despite the better final performance observed in the CMOM group, there were no statistical differences between this therapeutic group and the others, nor additional statistical differences between any therapeutic groups.

#### 3.2.2. Nociceptive Function

The WRL value, in seconds, recorded over 20 weeks can be seen in Figure 6. The complete WRL values can be found in Appendix A, and the statistical differences observed in T20 in Appendix A.

After neurotmesis injury, a general increase in WRL time was observed in all therapeutic groups. At week 1 after surgery, in all groups, a WRL time equal to or higher than the maximum considered value (12 s) was observed, and in all cases, it was necessary to manually remove the paw in contact with the heating plate to avoid skin thermal burns. At this timepoint, no statistical differences were recorded between the study groups, but between them and the UC group (*p* < 0.0001). From week 2 onwards, a progressive decrease in the WRL value was observed in all therapeutic groups. The EtE group showed a less pronounced decrease over time, with this group having the worst result at week 20, showing statistically significant differences with all other therapeutic groups and with UC (*p* < 0.0001). The CMOEC group also showed a less pronounced decrease in the WRL value between weeks 6 and 12, but the final value observed is identical to that of the other therapeutic groups. At 20 weeks, the lowest WRL value was observed in the CMOM group, but there were no additional statistical differences between the therapeutic groups and between these and UC.

#### 3.2.3. Walking Track Analysis

The SFI values recorded over the 20 weeks of the study are shown in Figure 7. The complete SFI values can be found in Appendix A, and the statistical differences observed in T20 in Appendix A.

After the neurotmesis injury, the functional consequences of the sciatic nerve injury translated into a significant impairment of the hind limb in all intervened groups. One week after surgery, with low SFI values being observed in all groups, it was in the EtE group where the functional index had the worst value, a position that the group maintains until the end of the 20 weeks of study. At this timepoint, not only does the EtE group show statistical differences with all other therapeutic groups (*p* < 0.0001 with CMOEC, *p* = 0.0492 with CMOM, *p* = 0.0003 with ECMOM, *p* = 0.0005 ECMOEC), but the UC shows significant differences with all remaining groups (*p* < 0.0001). Right after week 1, a progressive improvement in SFI is observed in all therapeutic groups, which remains consistent until week 20. At the final timepoint, the CMOM group is the one with the best functional index values, with statistical differences with the EtE group (*p* = 0.0016). The UC group showed statistical differences with EtE (*p* < 0.0001) and ECMOEC (*p* = 0.0097).

SSI values recorded over the 20 weeks of the study are shown in Figure 8. The complete SSI values can be found in Appendix A, and the statistical differences observed in T20 in Appendix A.

As expected, the evolution of SSI values is identical to that of SFI. The EtE group was once again the one with the worst performance throughout the entire study period, but at 20 weeks, it only showed statistical differences with the UC group (*p* = 0.0004). The remaining groups do not present statistical differences between them, and the UC group presented additional differences with the ECMOEC group (*p* = 0.0039). Even considering the absence of statistical differences, the CMOM group was again the one with the best performance at 20 weeks.

Table 1 includes a qualitative description of the therapeutic groups’ performance in the different functional assessment tests performed in vivo. In general, it is possible to perceive that the EtE group presented a markedly inferior performance when compared to the groups that received CM. The results of the groups that got CM from MSCs and OECs are identical in all parameters, although in the evaluation of the gait pattern, the ECMOEC group showed a slightly lower performance.

### 3.3. Stereological and Histomorphometric Analysis

#### 3.3.1. Nerve Stereological Analysis

The results obtained after the stereological evaluation are shown in Figure 9. The respective stereological images can be found in Figure 10. The total stereological results can be consulted in Appendix A and the respective statistical differences in Appendix A.

After the 20 weeks of study, all the nerves collected in the different therapeutic groups show stereological characteristics indicative of nerve fiber regeneration. In direct comparison with the nerves of the UC group, the nerves of the treated groups show microfasciculation phenomena, with a higher density of fibers, axons and fibers of smaller diameter, and a thinner myelin sheath. In all parameters studied, statistically significant differences were observed between the UC group and most of the therapeutic groups. In the parameters number and density of fibers, the EtE group presented the highest values (17,423 ± 2217 fibers and a density of 30,072 ± 5443 fibers/mm^2^, respectively). In terms of density, statistical differences were observed between UC and all other groups (*p* < 0.0001 with EtE and CMOEC, *p* = 0.0006 with ECMOM, *p* = 0.0004 with ECMOEC, *p* = 0.0017 with CMOM) and between EtE and CMOM (*p* = 0.0009), ECMOM (*p* = 0.0086), and ECMOEC (*p* = 0.0128). Differences were also identified between CMOM and CMOEC (*p* = 0.0115). For the number of fibers, statistical differences were observed between the EtE and UC (*p* = 0.0199) and CMOEC (*p* = 0.0006) groups. For the axonal and fiber diameter parameters, the EtE group presented the lowest value in both cases (2.374 ± 0.05819 μm and 3.768 ± 0.09515, respectively) and the highest value among the therapeutic groups was observed in the CMOM group in the axonal diameter (2.862 ± 0.1884 μm) and in the ECMOM group in the fiber diameter (4.342 ± 0.2529 μm). In both parameters, statistically significant differences were observed between UC and all other groups (*p* < 0.0001). The highest myelin thickness was observed in the ECMOM (0.7080 ± 0.02083 μm) group and the lowest in the CMOEC group (0.5700 ± 0.04889 μm). Additionally, in this case, statistically significant differences were identified between the control group and the remaining groups (*p* < 0.0001). The largest final cross-sectional area is observed in the CMOM group (0.6258 ± 0.02711 mm^2^) and the smallest in the CMOEC group (0.3890 ± 0.03940), with the UC group showing statistical differences with all other groups (*p* = 0.0006 with ECMOEC, *p* = 0.0001 with CMOEC, *p* = 0.0012 with ECMOM, *p* = 0.0060 with CMOM and *p* = 0.0041 with EtE). Finally, regarding the g-ratio, the highest value was that of the CMOEC group (0.6900 ± 0.0200) and statistical differences were observed between UC and CMOEC (*p* = 0.0032) and ECMOEC (*p* = 0.0292) and between EtE and CMOEC (*p* = 0.0047) and ECMOEC (*p* = 0.0477).

#### 3.3.2. Histomorphometric Muscle Analysis

The percentage of muscle mass lost in the cranial tibialis muscles of each therapeutic group, compared to the contralateral control muscle, is shown in Figure 11. The total muscle mass loss results can be consulted in Appendix A. All cranial tibial muscles associated with sciatic nerve injuries have a lower final mass than the contralateral healthy muscle. The group where a lower muscle loss was observed is the CMOM group, with an average loss of 29.14 ± 7.06%. In apposition, the ECMOEC group was the one where the final muscle mass is lower, with an average loss of 57.84 ± 14.53%.

The results of the histomorphometric evaluation of the cranial tibial muscle, namely the values of fiber area and minimum Feret’s diameter, are shown in Figure 12. The statistical differences can be observed in Appendix A.

In the fiber area evaluation, the highest final values were recorded in the groups CMOM (2700.57 ± 632.7 μm^2^) and ECMOM (2713.07 ± 628.5 μm^2^), followed by the EtE group (2510.91 ± 1137 μm^2^). In fact, the CMOM and ECMOM groups present final values slightly higher than the control UC group, but with no identified statistical differences. The UC group does show statistical differences with the CMOEC and ECMOEC groups (*p* < 0.0001), these being the groups with the lowest final fiber area values (2192.9 ± 753.8 μm^2^ and 2102.06 ± 757.1 μm^2^ respectively). Additional statistical differences were identified between EtE and the remaining therapeutic groups (*p* < 0.0001), between CMOM/ECMOM and CMOEC/ECMOEC (*p* < 0.0001) and between CMOEC and ECMOEC (*p* = 0.0051). For the minimum Feret’s angle parameter, the EtE group presented a higher final value (47.42 ± 11.15 μm). The groups CMOM (45.80 ± 10.62 μm), CMOEC (45.29 ± 10.06 μm), and ECMOEC (44.82 ± 4.26 μm) followed, with very close values and without statistical differences between them. The lowest value in this parameter was recorded in the ECMOM group (43.23 ± 5.228 μm). Statistical differences were identified between the UC group and all therapeutic groups (*p* < 0.0001), between the EtE group and all the remaining therapeutic groups (*p* < 0.0001), between CMOM and ECMOM (*p* < 0.0001), and between ECMOM and CMOEC/ECMOEC (*p* < 0.0001).

## 4. Discussion

In recent years, there has been increasing evidence that MSC secretome may have a neuroprotective and neurotrophic effect. In fact, the different studies in which the secretome and CM of MSCs have already been characterized have demonstrated the presence of various biomarkers and factors with angiogenic potential, growth factors, cytokines, and, more specifically, neurotrophic factors. Thus, the application of the secretome as a cell-free therapeutic approach may have benefits in promoting nerve regeneration, with clinical effects that involve immunomodulation at the site of injury, pro-vasculogenic effects, modulation of the different phases of Wallerian degeneration, increase in the thickness of the myelin sheaths, increase in the number and organization of nerve fibers, and decrease in the phenomena of fibrosis and exuberant scarring [22]. In addition to its potential direct effects on the promotion of peripheral nerve regeneration, the use of secretome as a therapeutic option has several technical advantages over the direct administration of cells: there are no concerns or limitations associated with the survival of MSCs after transplantation; the absence of proteins present on the cell surface decreases the immunogenic potential and the rejections after administration; the secretome can be stored for long periods of time as a ready-to-use and off-the-shelf product, and can be used when necessary without the need to expand large amounts of cells that can undergo phenotypic changes and lose therapeutic potential along the passages, in addition to not requiring the use of potentially toxic cryoprotectants; large amounts of secretome can be faster produced in bioreactors, increasing the efficiency of the process; the secretome can be modified and adapted for each clinical need [8,22,34]. Thus, there is a great clinical and scientific interest in developing regenerative therapies based on the administration of the cell secretome, as a whole or based on its constituent components, such as microvesicles and exosomes [35].

OM-MSCs had already been subjected to some studies where their secretome was analyzed, in which it was possible to identify the production and secretion of molecules related to angiogenesis, cell growth and migration, immunomodulation, and neurotrophy [12,36]. In this work, the CM of the rat OM-MSCs obtained in two cell passages after 24 and 48 h of conditioning was subjected to a new analysis for further characterization of its therapeutic and multifactorial potential. The aim of this analysis was not only to identify the components of the CM, but also to understand the influence of the passage and the conditioning time in its constitution. At the same time, and in order to compare the constitution of the secretome and to later compare the therapeutic efficacy in vivo, the CM of OECs was also analyzed under the same conditions. OECs are special glial cells, specifically associated with the olfactory nerve, and that due to their active participation in the regeneration process in the olfactory nervous system, have been explored as potential therapeutic agents for use in neural repair [37]. Although thus far they have largely been applied to study their effectiveness in promoting central nervous system regeneration [38], OECs have also been used and demonstrated efficacy in promoting peripheral nerve regeneration [39]. Analyses of its secretome also revealed the presence of several neurotrophic factors [38,40].

Schwann cells play an essential role in regulating the phenomena of axonal extension, protein synthesis, and remyelination, and it is now well established that the secretome of these cells plays an essential role in modulating Wallerian degeneration and sustaining axonal regeneration [41]. Thus, it is essential that the secretome selected as a therapeutic approach has in its constitution a set of biofactors also identified in the secretome of Schwann cells and that are known to actively participate in the regenerative process. Most of the cytokines and chemokines involved in this process have the function of promoting a macrophage chemoattraction that guarantees the cleaning of cellular debris and myelin fragments prior to the regenerative phase of Wallerian degeneration [42]. Among the most important factors involved in this mechanism are IL-1α, IL-1β, MCP-1, MIP-1α, and IL-6 [43]. The chemokines MCP-1 and MIP-1α play a very important role in the recruitment of macrophages, and IL-1α, IL-1β and MCP-1 modulate the phagocytic capacity of these cells in injured nerves through the stimulation of phospholipase A2 [43,44]. IL-1β also actively stimulates Schwann cell proliferation [45]. IL-6 is secreted by Schwann cells a few hours after nerve injury, which indicates its importance in the acute phase of regeneration and in the local inflammatory response. Simultaneously, some anti-inflammatory cytokines such as IL-10 are also upregulated at this stage, probably contributing to the balance in the immune response and preventing deleterious inflammatory reactions [41]. Fractalkine participates in a signaling pathway in neuron-to-microglia communication and is related to the development of neuropathic pain phenomena after nerve injury [46], and the cytokine GRO/KC shows upregulation in dorsal root ganglion after nerve injury, appearing to have a pro-nociceptive effect by increasing neuronal excitability in small diameter sensory neurons [47]. Along with MCP-1 and MIP-1α, RANTES also seems to have an important role in the recruitment of macrophages, but having a modulating effect on the secretion of proinflammatory cytokine proteins and regulating the inflammatory microenvironment at the site of injury [48,49]. The presence of IP-10 in the dorsal root ganglia after nerve injury seems to be associated with the development of neuropathic pain, but its upregulation in the peripheral nerve may also contribute to debris clearance in the injured nerve stumps [50]. LIX promotes cell survival and axonal regeneration [51]. VEGF, as a stimulating factor of angiogenesis and vasculogenesis, actively participates in the regenerative process, namely ensuring irrigation of the regeneration site and increasing Schwann cell migration [52]. EGF is a growth factor whose mRNA concentration and protein levels increase after nerve injury, both in the proximal and distal nerve stumps [53]. G-CSF is able to accelerate and intensify the regeneration of injured nerves, promoting a recovery of motor nerve conduction velocity and amplitude and a preservation of α-motoneurons [54]. Leptin is a hormone involved in the regulation of neuronal survival and development and has already been shown to have neuroprotective effects in cases of demyelination [55]. Some interleukins, such as IL-17 and IL-18, or factors such as MIP-2 seem to play an active role in the development of neuropathic pain phenomena and their relationship with the nerve regeneration process is poorly understood [56,57,58]. IFN-γ, as a classic pro-inflammatory cytokine, by stimulating M1 macrophage activation and consequent production of pro-inflammatory cytokines and high levels of oxidative metabolites, may have a harmful effect on the peripheral nerve regenerative process [59].

In this work, two passages were used in each cell type to evaluate their CM. It is known that the increase in the number of passages is associated with both phenotypic and behavioral changes in cultured MSCs, mainly due to senescence phenomena and chromosomal changes that progressively accumulate over time [60]. Previously, a tendency was identified for OM-MSCs to lose some of their multipotency capacity in higher passages, although with evidence of showing a greater tendency for specific differentiations over time [12]. In this way, to understand if the influence of the passages also extends to the production of biofactors after conditioning, two different passages were used for direct comparison and selection of the ideal passage for later assays. The CM analysis revealed the production of several of the biofactors described above by both cells. Of the 27 biofactors searched, 10 were identified in the OM-MSCs CM and 19 in the OECs CM. Comparing the variation in the concentration of markers along the passages, it is possible to perceive that a large number of biomarkers are only identified in higher passages, P7, and whenever they are identified in P4 and P7, the concentration in P7 is higher. Likewise, the concentration of markers also increases within the same passage when the conditioning time is longer, with the highest concentrations observed in 48 h conditioning. In this way, and considering the cellular performance of the OM-MSCs characterized previously [12], cells in P6 were selected for the production of CM to be used therapeutically in the later phases of the work, with conditioning to be maintained for 48 h.

Although the complex relationship of cytokines, growth factors, and neurotrophic factors that intervene in the peripheral nerve regeneration process is still not completely understood, it is unequivocal that after conditioning, both OM-MSCs and OECs are able to produce factors capable of participating actively in the different processes and phases of nerve regeneration. This finding reinforces the potential for the secretome of these cells to be used as a replacement for them as a therapeutic factor. Although in the CM of the OECs, a greater number of biofactors were identified, it is important to note that in the OM-MSCs CM, cytokines and interleukins were identified which are described as having a central role in the regenerative process and in the action of Schwann cells, such as MCP-1, IL-1β, IL-10, or VEGF. It is also interesting to note that biomarkers were identified in the secretome of OECs that are thought to be associated with the development of neuropathic pain phenomena, such as IL-17A, IL-18, and MIP-2, which is not the case in OM-MSCs CM. The lack of research on specific neurotrophic factors that are known to play a central role in nerve regeneration, such as the neurotrophin nerve growth factor, derived neurotrophic factor, or glia derived neurotrophic factor, does not allow a comparison of the secretomes to be made regarding their direct action on the neural regenerative process, and this targeted analysis should be developed in future works.

After the surgical intervention, as expected, all animals showed significant motor deficits, resulting from the complete transection of the sciatic nerve and interruption of the efferent impulses to the effector muscles. After the immediate post-operative period, a slight improvement in motor function was observed, values that remained stable until the tenth week. After this timepoint, there was a significant improvement in motor performance in all therapeutic groups, indicative of a greater regeneration of motor fibers in this period. At 20 weeks post-surgery, all therapeutic groups presented a value of motor deficit significantly lower than that recorded immediately after surgery, which indicates that all therapeutic options were capable of promoting motor functional recovery. The CMOM group was the one in which the lowest value of motor deficit was observed in T20, standing out from the remaining groups that presented similar final values. The EtE group, which presented the highest values of motor deficit in the immediate post-surgical period, approached the performance of the remaining groups over time, ending up with no statistical differences with them. Despite the evident improvements in motor function, none of the therapeutic groups recorded final performance values close to those of the control group, maintaining significant statistical differences at the end of the study period. On the other hand, there were no statistical differences between the final values of motor performance of the therapeutic groups under study, not even between the CMOM group and the remaining ones. Despite the motor performance evolution following a pattern identical to that observed previously after the application of OM-MSCs and Reaxon^®^ NGCs, the EPT values in T20 are slightly higher with the application of CM, indicating a less effective return to motor functionality. In the first study, more evident differences were identified between the EtE group and the remaining ones [17].

The WRL times recorded in all animals after the surgical intervention corresponded to the maximum value considered for this test (12 s), a direct consequence of the neurotmesis lesion and the loss of transport of nociceptive information from the peripheral thermal receptors. Between the second and fourth week after surgery, a slight decrease in WRL time was observed in all therapeutic groups, indicating the beginning of the sensory fiber regeneration and recovery of thermal sensitivity. From the fourth week onwards, the decrease in WRL times became more evident, and this functional improvement was consistently maintained throughout the remaining period of the study. Despite also registering progressive improvements over time, the EtE group was the one in which they were less evident, with this group ending up with the highest WRL times in T20. Among the remaining therapeutic groups, the CMOM group presented the lowest final WRL time. The CMOEC group had the second worst final performance. The functional improvements observed were sufficient for the final performance to be identical to that observed in the UC group, with no statistical differences between the control group and the therapeutic ones. The exception is made to the EtE group which, in addition to the worst result in T20, presented significant statistical differences with all the remaining groups. Sensory evaluation results and final WRL values appear to be identical after application of OM-MSCs or their CM, with marked differences between the EtE group and the remaining ones in both cases [17].

The results observed in the SFI and SSI tests are identical, as expected. After the surgical intervention, the indexes in both tests dropped to values close to the minimum considered for neurotmesis lesions. From the first week after surgery, an improvement in the SFI and SSI values was observed, with the indexes increasing progressively over time until the last timepoint. At T20, the CMOM group was the one that presented the highest values, and along with the other therapeutic groups, no statistical differences were observed with the UC group, which demonstrates the good functional recovery that resulted in evident improvements in the gait pattern. The group with the worst result was EtE, having remained in this position from the post-surgical period until T20, and showing statistical differences with UC in this last timepoint. The ECMOEC group was the second with the worst performance, so that in the SSI test, it presented statistical differences with the UC group at the final timepoint. The evolution in the gait pattern over the 20 weeks and the final results of SFI and SSI are identical after the application of OM-MSCs or CM, with a markedly superior performance compared to the EtE suture in both cases [17].

Overall, regarding the functional assessment, the application of all the therapeutic options under study resulted in an improvement in motor function, nociception, and gait pattern, with the CMOM group, which received OM-MSCs CM and a Reaxon^®^ NGC, to show the best results in all assays. Despite this improvement, in the EPT test, there was not a sufficient functional recovery for the final performance to be comparable to that recorded in the control group, a situation that was observed in the evaluation of sensory recovery and gait pattern. The EtE group had the worst results in the WRL, SFI, and SSI trials, but in the EPT, it had similar results to the other therapeutic groups. In a direct comparison between the applications of the two types of CM, there seems to be an overall superior functional performance in the OM-MSCs CM compared to the OECs CM, although these differences do not translate statistically. In the comparison between the direct application of OM-MSCs or its CM, there are no clear differences between the two therapeutic strategies, with both promoting motor, sensory, and behavioral functional recovery. The direct application of cells, however, seems to promote a better recovery of motor function than the corresponding CM [17].

Stereological evaluation performed at T20 revealed the presence of microfasciculation phenomena in all therapeutic groups, with the analyzed nerves presenting a high number and density of fibers, fibers and axons of a small diameter, and thin myelin sheaths, observations expected in injured peripheral nerves that are undergoing a regeneration process. Microfasciculation is also easily identifiable in Figure 10, where the histomorphometric differences between the UC group and the therapeutic groups are evident. As the regenerative process progresses, it is expected that microfasciculation, typical of the post-surgical period, will reverse and there will be an increase in fiber and axon diameter parallel to a decrease in fiber number and density. In T20, it is possible to observe that the groups treated with OM-MSCs CM have the lowest fiber density, while the group that received OECs CM have the lowest number of fibers. In both parameters, the EtE group has the highest value, being, therefore, the group where the regenerative process is more delayed at the time of euthanasia. Regarding density, however, marked statistical differences are observed between the therapeutic groups and UC, and none of the therapeutic options resulted in a density close to that observed in the control group. Nerve fiber diameter is one of the most important parameters in stereological assessment as it relates myelin sheath thickness to axonal diameter and is a determining factor in nerve conduction velocity [61]. The animals that received OM-MSCs CM had the highest values in the three parameters, *d*, *D*, and *M*. The EtE group had the lowest values of axonal diameter but had a higher myelin thickness than the CMOEC and ECMOEC groups. In any case, in the three parameters, significant statistical differences are observed between all the therapeutic groups and the UC, and there are no differences between the therapeutic groups under study, which indicates that after 20 weeks, there is still no histomorphometric reorganization capable of match the healthy nerve. The g-ratio is an axonal geometric constant that relates the degree of myelination and their cross-sectional size. By determining the ratio between the inner axonal diameter and the total diameter of the nerve fiber (*d/D*), it is possible to determine the quality of axonal myelination. Thus, together with the axonal diameter, the g-ratio is a determining factor in the velocity of neuronal conduction [62]. The optimal g-ratio value should be between 0.6 and 0.75 [63,64]. Variations in the thickness of the myelin sheath, even if slight, can translate into g-ratios above or below this ideal range, resulting in disturbances in conduction velocity and consequent changes in motor skills and sensory integration [65]. The g-ratio values in all the therapeutic groups of this work are within the ideal range, with the EtE group and those that received OM-MSCs CM presenting the values closer to UC. The groups that received OECs CM showed higher g-ratio values and statistical differences with UC. The outcomes resulting from the stereological evaluation of the intervened sciatic nerves reveal that all the therapeutic options promoted phenomena of nerve regeneration, good axonal myelination, and potential velocity of neuronal conduction identical to the healthy nerve, which, moreover, justifies the results observed in the functional evaluation and recovery in motor performance, nociception, and gait behavior. Although in the density and number of fibers the groups that received OM-MSCs CM presented better results and the EtE group the worst performance, the absence of statistical differences between the therapeutic groups and the marked differences with the UC group in the remaining parameters does not allow to establish any therapeutic option as unequivocally more beneficial. It seems, however, that there is a tendency for the worst performance after the application of OECs CM compared to the application of OM-MSCs CM. Compared with the results obtained after application of cells [17], the administration of CM leads to lower values of density and number of fibers and higher values of axonal diameter, myelin thickness, and fiber diameter, with identical values of g-ratio, seeming to promote a better histomorphometric reorganization in T20.

One of the direct consequences of peripheral nerve damage is the chronic constriction of efferent skeletal muscles, which can translate into debilitating phenomena, such as neuropathic pain, changes in motor performance, and neurogenic atrophy [66]. In addition to the application of OM-MSCs, resulting in the lowest percentage of muscle mass lost at 20 weeks, the CMOM and ECMOM groups also have the highest fiber area, without statistical differences compared to the UC group. The EtE group follows, with the CMOEC and ECMOEC groups presenting the worst performance in this parameter. In the determination of the Minimum Feret’s diameter, the EtE group has the best result, with the remaining groups presenting identical performances and the ECMOM group with slightly lower values. In this second case, however, all groups present statistically significant differences with the UC group. The results of the muscular histomorphometric evaluation seem to corroborate the good performance of the groups that received OM-MSCs CM, particularly the CMOM group, which was the one that presented the best results in all the functional evaluation tests and that also presented good stereological results. The EtE group also presents good results in this evaluation, which is in line with the functional results of this group in the EPT test as opposed to the less favorable results in the sensory evaluation and gait pattern tests. The performance of different therapeutic groups in functional assessments also confirms the importance of avoiding or reversing neurogenic muscle atrophy to ensure good performance in motor function and the ability to return to a gait pattern close to normal. Compared with the direct application of MSCs, the application of OM-MSCs CM resulted in a larger fiber area, with values of Minimum Feret’s diameter identical between the two therapeutic options [17].

As identified in previous works [17], the EtE group had the worst overall performance, excluding the histomorphometric evaluation of the cranial tibial muscle and consequent motor performance. These therapeutic results have been previously identified in other studies [67], even though the EtE suture continues to be a safe therapeutic option and considered as a gold standard, along with the application of autografts, in clinical scenarios of neurotmesis [68]. A possible explanation for this observation may be related to the difficulty in guaranteeing a perfect juxtaposition and alignment of the two nerve ends during the performance of the EtE suture in small nerves, in which it is difficult to establish a precise axon-to-axon and endoneurium-to-endoneurium alignment and reconnection. This, in turn, can facilitate the occurrence of misdirection and aberrant reconnections between sensory and motor fibers in the regenerating nerve, delaying and hindering functional recovery. On the other hand, the superior results observed when this technique is applied simultaneously with the administration of CM and with the application of Reaxon^®^ reinforce the fact that the application of NGCs promote good orientation of the nerve tops and avoid misdirection phenomena during nerve regeneration [69]. Additionally, the application of nerve wrap with NGCs after EtE and tension-free neurorrhaphy manages to overcome the possible limitations of isolated EtE suture and promote a better functional recovery and histomorphometric reorganization, as observed in the ECMOM and ECMOEC groups [70].

The promising in vivo results of this work can be a starting point for new therapeutic options based on the application of MSC secretome as cell-free alternatives to the administration of cellular products and their limitations. The CM analysis confirmed that different conditioning conditions can influence the characteristics of the cell secretome. After the identification of several biomarkers with an immunomodulatory function that can act in these lesions in an undifferentiated way, these results open doors for the manipulation of cell culture for the production of neurotrophic factors that could specifically act in nerve regeneration. The comparative evaluation of the application of MSCs and their CM demonstrates that both OM-MSCs and OM-MSCs CM have therapeutic potential to promote peripheral nerve regeneration, but despite all the known advantages associated with the use of the secretome, its performance does not unequivocally stand out on the results of functional recovery and histomorphometric reorganization. Our current misunderstanding of what are the ideal conditioning requirements for CM production may be limiting, to some extent, the full potential of cell secretions as a therapeutic method. The use in future works of cells in other passages, conditioned for longer or shorter periods of time, or even the use of alternative conditioning methodologies such as priming, use of alternative culture media, or hypoxic conditions may reveal new details about the potential of this technique and maximize the therapeutic efficacy of the OM-MSCs CM as an alternative to the use of cells and its well described disadvantages. There were slight differences between the CM of OM-MSCs and OECs after in vivo application, although the CM obtained from the OECs was richer in biomarkers than that obtained from OM-MSCs. These results allow a combined use of OECs secretome and OM-MSCs, maximizing the therapeutic advantages of each method. This approach has been used previously with good results, with the secretome of OECs having a positive effect on the metabolic activity and proliferation of MSCs from different origins [67]. Considering the anatomical and functional relationship of these two types of cells in the OM, it is expected that this interaction could be even more beneficial in this case. Despite the promising results observed after the application of CM in vivo, the fact that a limited number of animals were used in each therapeutic group does not allow to amplify and generalize the conclusions discussed here, requiring their validation in subsequent assays. In addition, the already repeatedly identified limited equivalence of results obtained in the rat model for clinical application in human or veterinary medicine [71] also requires further testing of these therapeutic applications in more complex animal models that better mimic real clinical scenarios and function as intermediate bridges in a translational perspective [72]. Furthermore, although different important dimensions in the functional assessment of nerve regeneration have been explored in this work, new functional tests may be used to further determine the ability of the studied therapeutic options to promote peripheral nerve regeneration after injury, namely kinematic analysis, nerve conductivity tests, or tracing methods that were not considered here [30].

## 5. Conclusions and Further Directions

The use of MSCs, their secretion products, and biomaterials in promoting peripheral nerve regeneration after PNI has been a constant in research around regenerative medicine. Although with clear evidence of efficacy resulting from analyzes of the secretome of several types of MSCs, before this work, the secretome and CM of OM-MSCs had never been used to promote peripheral nerve regeneration, neither alone nor in combination with biomaterials, and this combinatory therapy proved to be effective. The CM analysis, through a multiplexing technique, allowed to simultaneously identify the presence of several biomarkers with immunomodulatory functions that could directly intervene in the degenerative and regenerative phases of Wallerian degeneration. The search for specific neurotrophic factors produced and secreted by these cells needs further research.

Functional and histomorphometric assessment confirmed that the use of CM and Reaxon^®^ NGCs ensured functional improvements throughout the study period with evidence of nerve regeneration after 20 weeks. Overall, the application of OM-MSCs CM resulted in better results in the functional tests, in the muscle histomorphometric evaluation, and in the stereological evaluation of the harvested sciatic nerves, followed by the performance of the OECs CM. The general performance of the CM application seems to be identical to the application of the cells themselves, with all the advantages associated with the use of cell-free techniques to strengthening this therapeutic option.

In conclusion, the results observed in this work make evident the potential of using OM-MSCs cell secretome to promote peripheral nerve regeneration, creating new therapeutic opportunities for the future. Even so, there are some irregularities resulting from the functional and histomorphometric evaluations that not only show the multifactorial complexity associated with nerve regeneration, but also create the need to carry out new studies to answer the doubts raised and allow the development of new therapeutic approaches based on the use of MSCs and their secretion products.

## Figures and Tables

**Figure 1 biomolecules-12-00818-f001:**
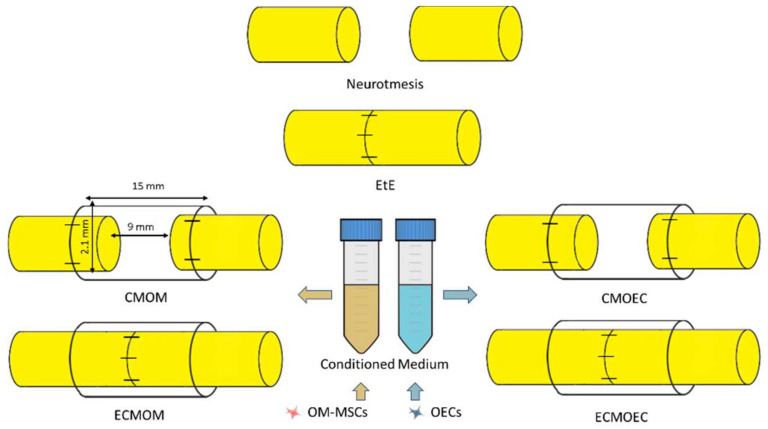
Experimental therapies applied to the sciatic nerve after neurotmesis injury. The applied NGCs are 15 mm long and have an internal diameter of 2.1 mm.

**Figure 2 biomolecules-12-00818-f002:**
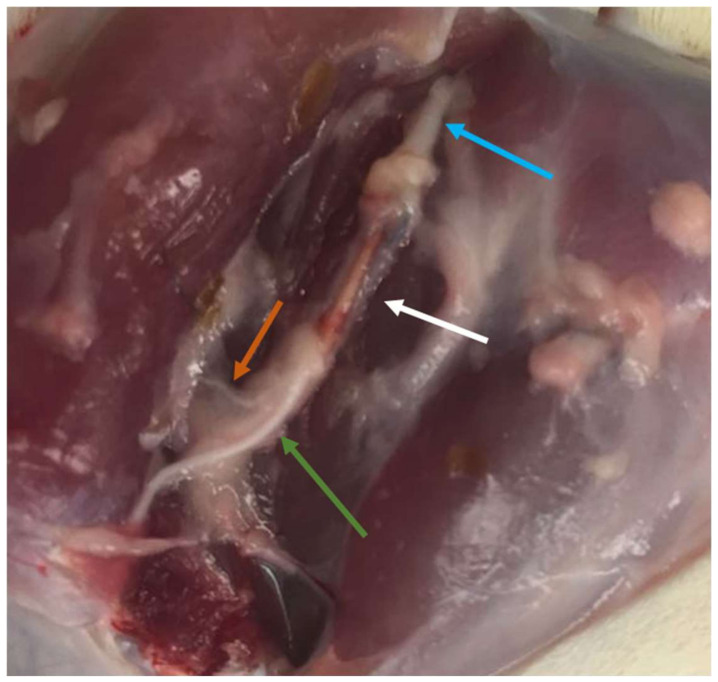
Exposure of the sciatic nerve (blue arrow) of the CMOM therapeutic group, 20 weeks after surgery. It is possible to observe the regenerated nerve filling all the lumen of the NGC (white arrow), with the two nerve tops connected, as well as the main branches of the sciatic nerve, the common peroneal nerve (green arrow), and the tibial nerve (orange arrow).

**Figure 3 biomolecules-12-00818-f003:**
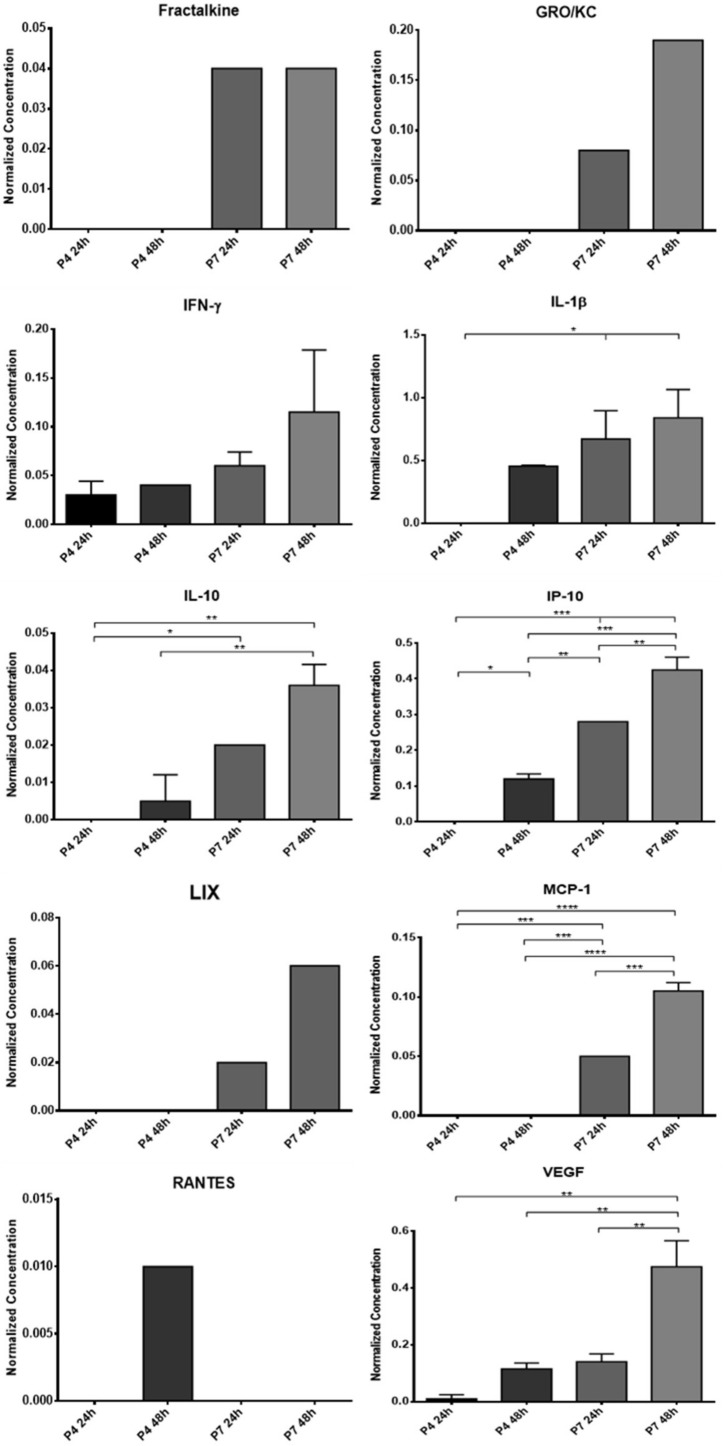
Normalized concentration of each biomarker in the conditioned medium of OM-MSCs in P4 and P7 (mean ± SEM). * corresponds to 0.01 ≤ *p* < 0.05, ** to 0.001 ≤ *p* < 0.01, *** to 0.0001 ≤ *p* < 0.001, and **** to *p* < 0.0001.

**Figure 4 biomolecules-12-00818-f004:**
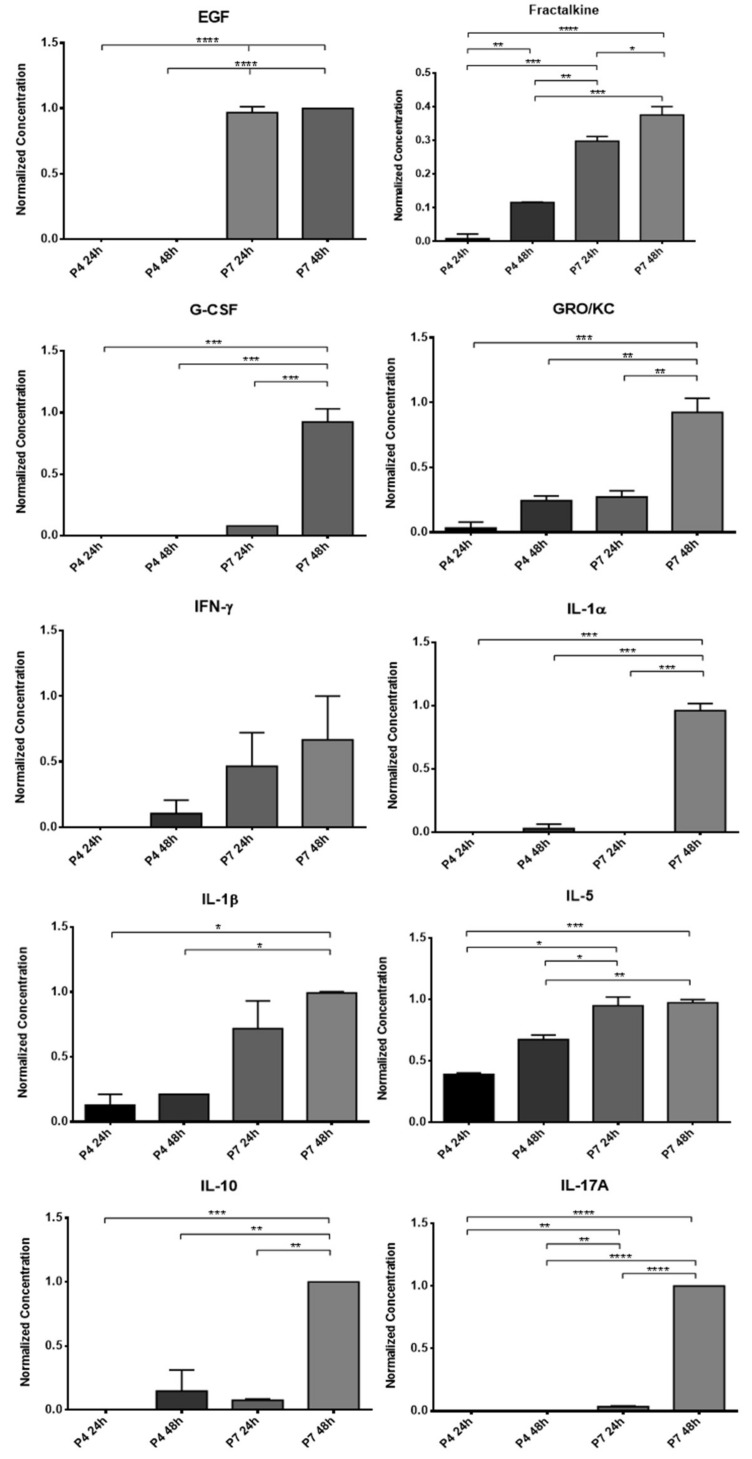
Normalized concentration of each biomarker in the conditioned medium of OECs in P4 and P7 (mean ± SEM). * corresponds to 0.01 ≤ *p* < 0.05, ** to 0.001 ≤ *p* < 0.01, *** to 0.0001 ≤ *p* < 0.001, and **** to *p* < 0.0001.

**Figure 5 biomolecules-12-00818-f005:**
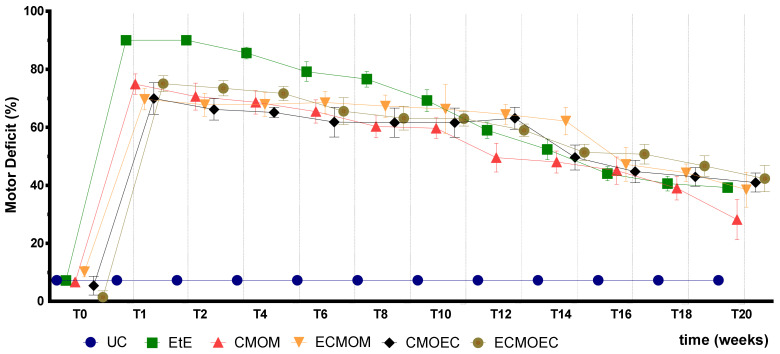
Values of motor deficit (%) over the 20 weeks of the recovery period (mean ± SEM). One week after the injury, all groups showed high percentages of motor deficit when compared to the UC group. Deficits gradually decreased over the study weeks, and at 20 weeks, the CMOM group showed the lowest value of motor deficit.

**Figure 6 biomolecules-12-00818-f006:**
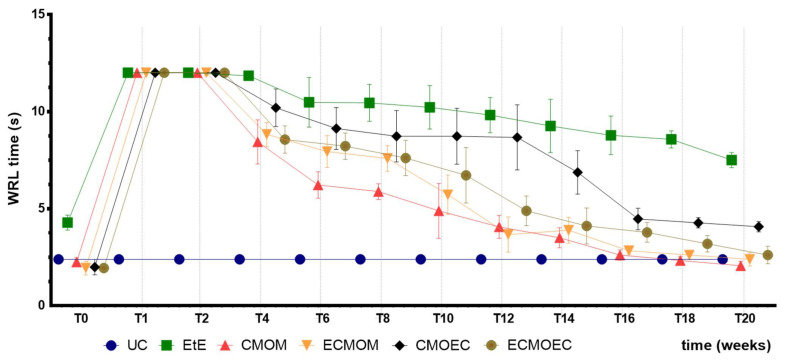
WRL values (s) over the 20-week recovery period (mean ± SEM). After neurotmesis injury, a general increase in WRL time was observed in all therapeutic groups. After week 2, a progressive decrease in the WRL value was observed in all groups, and at 20 weeks, the lowest WRL value was identified in the CMOM group.

**Figure 7 biomolecules-12-00818-f007:**
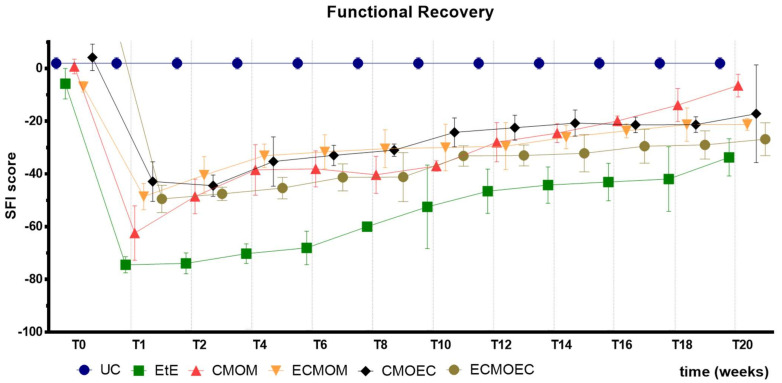
Functional assessment (SFI) over the 20-week recovery period (mean ± SEM). After the neurotmesis injury, a significant impairment of the hind limb was observed in all intervened groups. After week 1, a progressive improvement in SFI was observed in all therapeutic groups, and at the final timepoint, the CMOM group was the one with the best functional index values.

**Figure 8 biomolecules-12-00818-f008:**
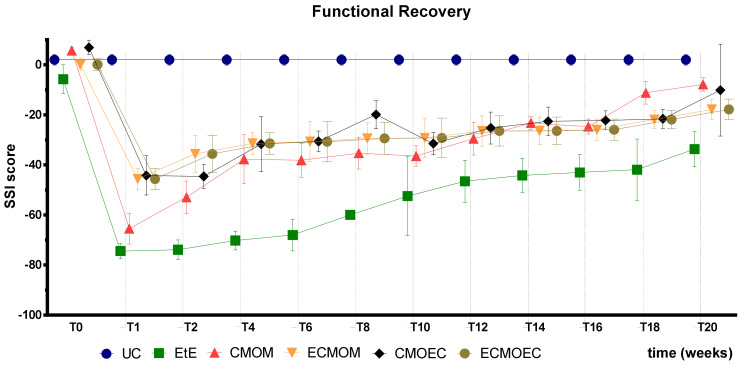
Functional assessment (SSI) over the 20-week recovery period (mean ± SEM). After the neurotmesis injury, a significant impairment of the hind limb was observed in all intervened groups. After week 1, a progressive improvement in SFI was observed in all therapeutic groups, and at the final timepoint, the CMOM group was the one with the best functional index values.

**Figure 9 biomolecules-12-00818-f009:**
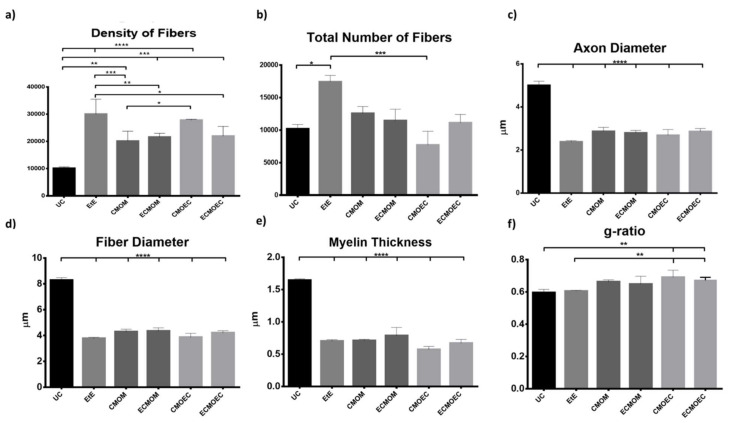
Results of the stereological assessment of sciatic nerve fibers 20 weeks after neurotmesis: (**a**) density of fibers; (**b**) total number of fibers; (**c**) axon diameter; (**d**) fiber diameter; (**e**) myelin thickness; (**f**) g-ratio (mean ± SEM)). * corresponds to 0.01 ≤ *p* < 0.05, ** to 0.001 ≤ *p* < 0.01, *** to 0.0001 ≤ *p* < 0.001, and **** to *p* < 0.0001.

**Figure 10 biomolecules-12-00818-f010:**
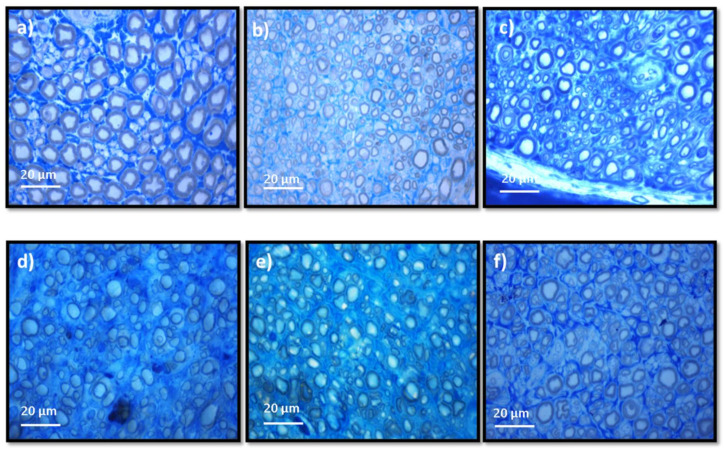
Light micrographs of toluidine blue-stained sciatic nerve semithin sections for the different groups: (**a**) UC; (**b**) EtE; (**c**) CMOM; (**d**) ECMOM; (**e**) CMOEC; (**f**) ECMOEC.

**Figure 11 biomolecules-12-00818-f011:**
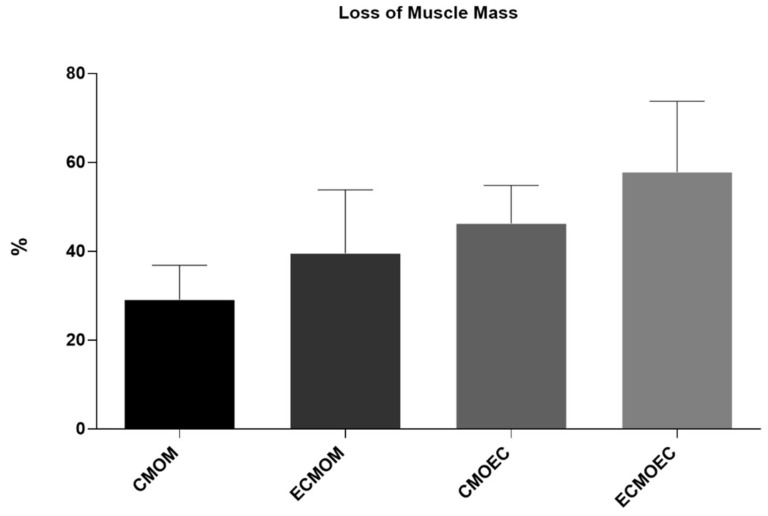
Percentage of muscle mass lost in each therapeutic group as a function of contralateral healthy muscle weight (mean ± SEM).

**Figure 12 biomolecules-12-00818-f012:**
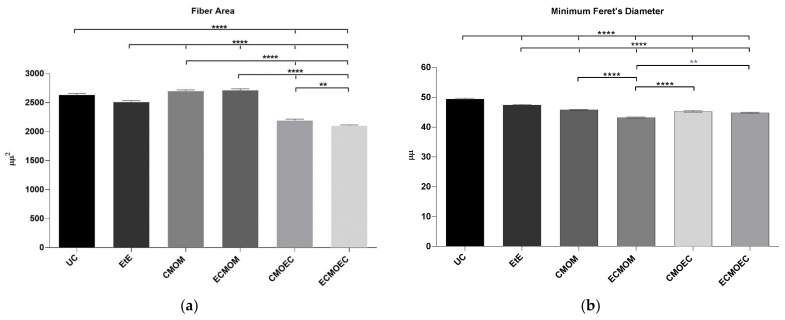
Histomorphometric analysis of cranial tibial muscle: (**a**) individual fiber area; (**b**) minimum Feret’s diameter of the muscle fibers (mean ± SEM). ** to 0.001 ≤ *p* < 0.01, and **** to *p* < 0.0001.

**Table 1 biomolecules-12-00818-t001:** Qualitative general classification of the performance of the therapeutic groups in the tests and essays carried out in vivo. The groups were classified according to the statistical differences observed between them and the UC group: **** (-); ***, **, * (+); and no statistical differences (++).

		EtE	CMOM	ECMOM	CMOEC	ECMOEC
**Functional Assessment**	EPT	-	-	-	-	-
	WRL	-	++	++	++	++
	SFI	-	++	++	++	+
	SSI	-	++	++	++	+

## Data Availability

The data that support the findings of this study are available from the corresponding author on request.

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
