# Peer review of "Effects of Olfactory Mucosa Stem/Stromal Cell and Olfactory Ensheating Cells Secretome on Peripheral Nerve Regeneration"

_biomolecules, 2022, doi:10.3390/biom12060818_

Round 1

Reviewer 1 Report

It is a quite complete well designed study on peripheral nerve regeneration in the rat after neurotmesis using chytosan conduits grafts filled with stromal stem cells from olfactory mucosa, olfactory ensheathing cells or their secretomes, the composition of which also has been achieved for a considerable number of biomarkers.

The methodology has been carried out properly assessing biochemical and functional tests as well as histological techniques. The authors claim to have also performed electron microscopy, but no description nor micrograph about ultrastructural findings is shown.

The discussion has reached too long extension, many parts of it not being actually discussion, especially the whole first paragraph which would rather be placed in the introduction. The introduction should include relevant references about the topic such as the following: Zurita M, Aguayo C, Bonilla C, Rodriguez A, Vaquero J. Perilesional intrathecal administration of autologous bone marrow stromal cells achieves functional improvement in pigs with chronic paraplegia. Cytotherapy, 15:1218-1227 (2013). Garrosa M, Alcalde I, Gamboa OL, Gayoso MJ. Peripheral nerve regeneration by means of a poly(lactide-co-glycolide) prosthesis seeded with Schwann cells. Anales. R. Acad. Med. Cir. Vallad. 48:159-173 (2011).

The authors compare their results with a previous paper by them, so they should explain why the former study showed more evident differences between EtE group and the remaining ones. It is interesting that EtE group, in which the nerve stumps are sutured directly, with no gap, show worse regeneration, especially compared with groups having a 9 mm gap between both stumps. A discussion comparing the results found in the groups with no gap with those in which a gap was created and grafted with the conduit should be added.

Furthermore, EtE group showed higher total number of fibers, but in table 1 it is shown a negative value meaning less regeneration, that is, functional assessment does not correlate with the histological findings, a matter that needs to be discussed trying to give an explanation. In addition, in lines 742-743 it is said that EtE group ended up with no statistical difference in most deficits. How does this reconcile with the great differences shown in table 1?

Figure 5 caption requires a small explanation of the results shown in the graph.

Figure 10 d) is too dark. It should be replaced for a better one.

The abbreviation for nerve guidance conduit (NGC) should appear in the abbreviation list.

Line 146: conditioned medium should be written before CM and CM should appear bold and in brackets.

Line 202: a space before (RANTES) is missing.

Lines 216 and 230: The number of 30 in UC group does not coincide with the sum of the animals in the remaining groups (28). Please correct it throughout the text, including the supplementary material.

Line 232: a space is missing before group 4.

Line 240 and 286: secondary instead of secundary.

Line 262: transection instead of tranception.

Line 318: following instead of follwing

Line 397: “d” and “D” are missing and should appear in their corresponding brackets.

Line 425: 0.001 instead of 0.0001.

Line 426: 0.0001 instead of 0.001.

Line 450: a space is missing before In.

Line 473: showed instead of show.

Lines 651 and 657: degeneration instead of Degeneration

Lines 720-729: This paragraph is inappropriate for the discussion. It has already been said in materials and methods.

Line 805: Figure 10 instead of Figure 16.

Line 857: diameter instead of Diameter.

Line 869: a full stop is missing before Compared.

Line 870: Ferret’s instead of ferret’s.

Lines 1054-1055: The reference appears in Chinese. Please correct it.

Author Response

Dear reviewer

Please consult the document in attachment with authors' notes and comments.

Reviewer 2 Report

Overall, this paper explored the secretome produced and collected from Olfactory Mucosa mesenchymal Stem cells and Olfactory Ensheating Cells that can contribute to promoting peripheral nerve regeneration. In this work, the authors analyze the conditioned medium which includes several factors with immunomodulatory functions has a beneficial impact on nerve regenerations including motor performance, sensory recovery, and gait patterns. This study would be beneficial for audiences who are engaged in studying in this field and provides a therapeutic potential applied in the treatment of peripheral nerves after injury. The experiment is well designed, and the data are well executed. 

Minor points:

1.    Remove double spaces in the entire paper.

2.    In tables S1 and S2, the authors missed “h” in P4 14h instead of P4 14

3.   In Figures 3 and 4, the authors didn’t explain what are the benefits to collect cells at these different passages (P4 and P7), what’s the biological mechanisms that make a difference between these two passages in terms of each biomarker expression?

4.     In figure 9a, it would be better to label the figure as “The density of Fibers” instead of “Density “which lacks of information.

Author Response

(The authors gave the same response as above.)
